# Comparative Analysis of a Family of Sliding Mode Observers under Real-Time Conditions for the Monitoring in the Bioethanol Production

Eduardo Alvarado-Santos [1], Juan L. Mata-Machuca [2,*], Pablo A. López-Pérez [3], Rubén A. Garrido-Moctezuma [4], Fermín Pérez-Guevara [1] and Ricardo Aguilar-López [1,*]

1   Department of Biotechnology and Bioengineering, CINVESTAV-IPN, San Pedro Zacatenco, Mexico City 07360, Mexico
2   Department of Advanced Technologies, Instituto Politécnico Nacional, UPIITA, Av. IPN 2580, Mexico City 07340, Mexico
3   Escuela Superior de Apan, Universidad Autónoma del Estado de Hidalgo, Carretera Apan-Calpulalpan, Km.8., Chimalpa 43900, Mexico
4   Automatic Control Department, CINVESTAV-IPN, San Pedro Zacatenco, Mexico City 07360, Mexico
*   Correspondence: jmatam@ipn.mx (J.L.M.-M.); raguilar@cinvestav.mx (R.A.-L.)

**Abstract:** Online monitoring of fermentation processes is a necessary task to determine concentrations of key biochemical compounds, diagnose faults in process operations, and implement feedback controllers. However, obtaining the signals of all-important variables in a real process is a task that may be difficult and expensive due to the lack of adequate sensors, or simply because some variables cannot be directly measured. From the above, a model-based approach such as state observers may be a viable alternative to solve the estimation problem. This work shows a comparative analysis of the real-time performance of a family of sliding-mode observers for reconstructing key variables in a batch bioreactor for fermentative ethanol production. These observers were selected for their robust performance under model uncertainties and finite-time estimation convergence. The selected sliding-mode observers were the first-order sliding mode observer, the proportional sliding mode observer, and the high-order sliding mode observer. For estimation purposes, a power law kinetic model for ethanol production by *Saccharomyces cerevisiae* was performed. A hybrid methodology allows the kinetic parameters to be adjusted, and an approach based on inference diagrams allows the observability of the model to be determined. The experimental results reported here show that the observers under analysis were robust to modeling errors and measurement noise. Moreover, the proportional sliding-mode observer was the algorithm that exhibited the best performance.

**Keywords:** state observers; sliding modes; real-time implementation; batch bioreactor; ethanol fermentation

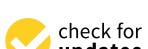



## 1. Introduction

In recent years, fermentation processes have become an industry of great economic importance, this has motivated scientists and engineers to seek new strategies to improve performance and reduce operating costs. A key question when optimizing this type of bioprocess is how to monitor all its critical variables online since this is a necessary task to bring the process to the desired state of operation [1]. However, the available instrumentation and sensors do not always cover all the necessary measurements or at least the necessary ones. The low availability of sensors in the market and their high costs, the presence of noise measurement, the operational politics of the bioreactors, and their intrinsic nonlinear behavior, are strong obstacles to bioreactor instrumentation [1,2].

Due to a growing need to optimize production processes and increase the quality of final products, engineers in the industry commonly see soft sensor (SSs) or virtual sensor (VSs), which are also called state observers, as a viable alternative for monitoring key

variables in the bioprocesses. Compared to expensive and relatively complex analytical techniques, these can provide reliable estimates online, require no maintenance, and are less expensive [3].

VSs are classified into two types: techniques based on historical data and model-based techniques. In the first case, the VS uses the historical values of the available measurements to build a model that allows the inference of the variables of interest. In this class of techniques, artificial neural networks, vector-supported machines, and regressive models stand out [4]. On the other hand, model-based techniques combine online measurements using physical sensors and an estimation algorithm based on an auxiliary system built using a model of the fermenter [1]. Figure 1 shows a general scheme of the implementation of virtual sensors based on a model implemented in a biological system.

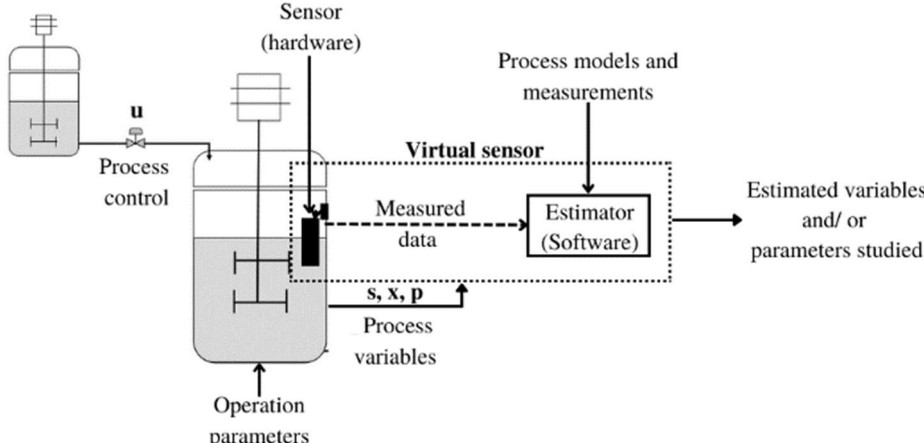

**Figure 1.** General scheme of the implementation of a virtual sensor in a bioprocess.

Some of the advantages of state observers are the following: (a) they require fewer computational resources and measurements compared to databased techniques, (b) the use of phenomenological models facilitates the interpretation of the results, and (c) they do not require a specific architecture for the processing or training stages [5].

There is literature in which the classic Luenberger and Kalman observers commonly stand out. They guarantee asymptotic convergence of the estimated states to the real states if the model is exactly known [6]; however, they may lack robustness in the face of parametric uncertainty and unmodelled terms, i.e., when the model used to build the observer does not exactly match the system under observation. Another problem is the unknown disturbances affecting the observed system. The classic observers do not consider these disturbances in their structure [7].

To address the above problems, there exist, in the literature, improved or extended versions of the Luenberger and Kalman observers. Another alternative is the sliding-mode observers, which have drawn attention due to their robust properties against external and internal uncertainties, modeling errors, and measurement noise. These observers force the trajectories of the states to an area called the sliding surface, where the estimation error converges to zero or close to zero. Moreover, they achieve finite time convergence [2,8].

A major factor limiting the real-time implementation of on-line monitoring techniques is the lack of suitable sensors. Table 1 shows the general panorama of the role that state observers play in bioprocesses. In many cases, numerical simulations permit the assessing of the performance of the observers, although there are also real-time implementations where the bioprocesses are already equipped with the necessary hardware and software for implementation [9–11]. For example, Petre et al. [12] developed an adaptive control-law design based on nonlinear estimation algorithms for unknown inputs and kinetics.

**Table 1.** Overview of the application of state observers in bioprocesses.

| Observer | Process | Measured Variables | Estimated Variables | References |
|---|---|---|---|---|
| Sliding-mode observer | Model of a stirred tank | Substrate concentration | Substrate consumption rate | [8] |
| Neural networks | Fermentation for yeast production by *Saccharomyces cerevisiae* | Substrate concentration volume of the medium | Biomass concentration Trehalose concentration | [9] |
| Asymptotic observer | Continuous fixed-bed anaerobic reactor used for wastewater treatment | Flow of $O_2$ in and out, volume, inlet fructose, and nitrogen concentration | Biomass concentration | [10] |
| Sliding-mode observer | Alcoholic fermentation process | Substrate and ethanol concentration | Influent substrate | [12] |
| Extended Kalman filter | Anaerobic digestion pilot plant | Methane flow outlet | Substrate concentration | [13] |
| Recursive Bayesian filter | Alcoholic fermentation by *Zymomonas mobilis M* | Substrate and product concentration | Biomass concentration | [14] |
| Super-twisting observer | Beer fermentation | Reducing sugars and ethanol via HPLC | Biomass concentration | [15] |
| Geometric observer | Yeast fermenter | Substrate concentration | Biomass concentration | [16] |
| Extended Luenberger observer | Anaerobic digestion model | Biomass concentration and volatile fatty acids | Concentrations of methane and carbon dioxide | [17] |
| Hybrid observer (linear and nonlinear Luenberger observer) | Biohydrogen production fermenter model | Concentrations of glucose and biomass | Production of hydrogen | [18] |

Another common practice is to carry out numerical simulations and validate the numerical results using the experimental data. For instance, Avilés et al. [18], designed an interval observer applied to a dark fermenter for the production of biohydrogen. The observer estimates the concentrations of glucose and biomass and the flux of hydrogen produced, and compares their results with offline experimental data.

This work studies the performance of three sliding mode observers for the real-time monitoring of the concentrations of substrate, ethanol, and $CO_2$ in a batch fermenter by the strain *Saccharomyces cerevisiae*, from the measurement of biomass. These observers are the sliding-mode observer, the proportional sliding-mode observer, and the high-order sliding-modes observer. The experimental configuration, the characteristics of the bioprocess and the implemented observation strategies are briefly exposed. A power-law kinetic model is proposed for the description of ethanol production. Numerical simulations are performed using a perturbed biomass concentration signal as the available measurement and the efficiency of each observation strategy is evaluated using the absolute integral error and squared integral error performance indices. Based on the results from the numerical simulations, online observation strategies were implemented, and finally the proposed approaches were validated with experimental data.

## 2. Materials and Methods

Figure 2 shows a general scheme of the methodology used in this research work. Firstly, an experimental database of the fermentation process-time evolution was generated with the main performance variables of substrate ($s$), biomass ($x$), ethanol ($E_t$) and carbon dioxide ($CO_2$). The above information was employed to develop a kinetic model to describe the dynamic behavior of the bioreactor. The adjustment of the kinetic parameters of the model was carried out using a hybrid parametric identification methodology. The prediction error of the model was minimized using the local optimization method to obtain the initial values of the parameters so that later a global optimization algorithm used those values to obtain the optimal values of the parameters [19]. Finally, the proposed kinetic model was validated through statistical correlation coefficients.

Once the bioreactor model was established and validated, the observability of the model was evaluated through a graphic methodology based on an inferential diagram. To show the ideal performance of the selected observers' structures, they are numerically implemented via dynamic simulations, to evaluate their performance, with the integral absolute error index (IAE) and the integral squared error index (ISE). The second phase of this methodology was the implementation in real-time, of the proposed sliding-mode observers. The performance of each observer was also evaluated using IAE and ISE where the estimation errors were defined as the difference between experimental values and the values of the state observers.

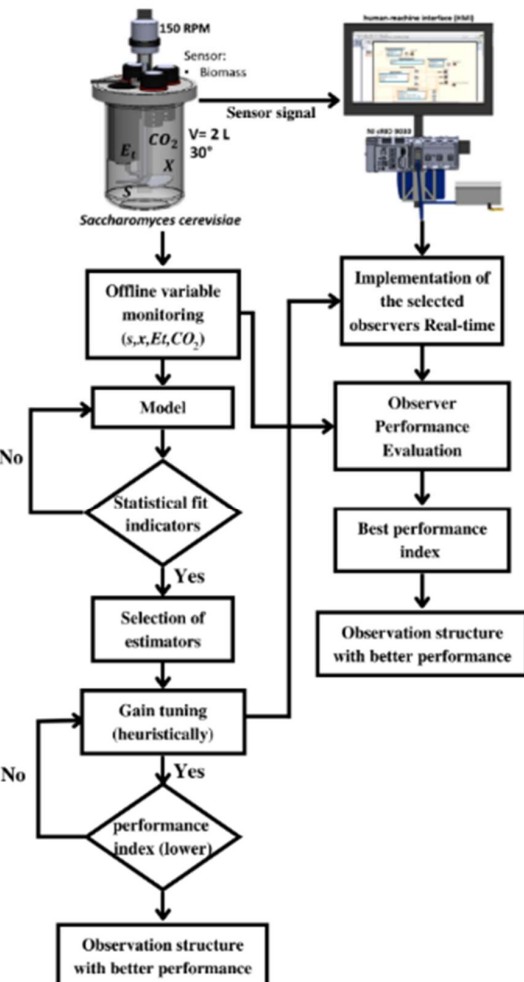

**Figure 2.** General scheme of the methodology.

## 2.1. Batch Fermenter

The scheme of the batch fermenter is shown in Figure 3. The reactor was an original design inspired by a stirred tank reactor to ensure a homogeneous mixture in the tank. The tank was instrumented with a low-cost turbidity probe.

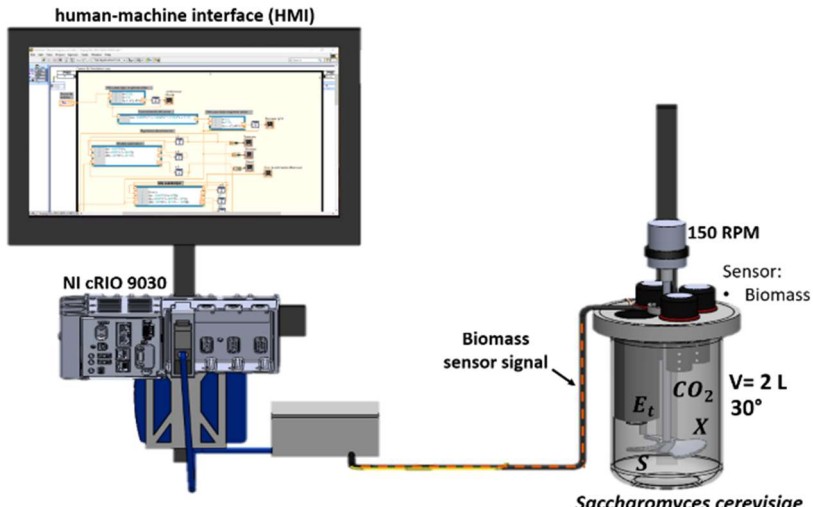

**Figure 3.** Schematic of the experimental setup in batch operation.

The following components comprise the experimental setup:

(a)     NI CRIO-9030: high-performance real-time controller with a reconfigurable FPGA chassis, using a 1.33 GHz dual-core Intel Atom chip;
(b)     NI 9381: 24-bit analog input module for the cRIO real-time embedded system;
(c)     Turbidity sensor (used to determine the biomass concentration);
(d)     Tank with a capacity of 2 L;
(e)     12 V voltage source;
(f)     LED monitor as a user interface.

The fermentation was carried out with an inoculum at 0.1 g L$^{-1}$ of *Saccharomyces cerevisiae*, with 1 L of sterile YM medium. The initial substrate concentration was adjusted $S_0$ = 54 g/L. The incubation temperature was 30 $\pm$ 3 °C and agitation was 150 rpm [20]. The fermentation time was 24 h.

The optical-density-determined cell-count spectrophotometric method (at a wavelength of 230 nm), the dry weight method, and the GT-TSW-30 probe to perform online measurements [21]. Total reducing sugars were estimated using the dinitrosalicylic acid (DNS) colorimetric method adapted from [22]. Reducing sugars were calculated using the regression equation (standard curve with glucose (1 mg mL$^{-1}$)). Ethanol was determined by chromatography. The standard curve was obtained with HPLC-grade ethanol (Sigma-Aldrich). A Varian CP-9002 gas chromatograph with a flame ionization detector equipped with a ZB-FFAP column was used. The concentration of $CO_2$ was monitored with the $CO_2$-BTA Vernier probe; this probe monitored the amount of infrared radiation absorbed by carbon dioxide molecules. A $CO_2$ gas sensor monitored carbon dioxide production as the yeast consumed different initial concentrations of glucose [23].

*2.2. Proposed Kinetic Model*

Mathematical models allow the essential characteristics of the phenomena that occur within bioprocesses to be described. In addition, they play a key role for model-based observation algorithms since the efficiency of the observer depends on the ability of the model to emulate the bioprocess dynamics [1].

Kinetic models provide fundamental information for the design and optimization of bioprocesses. To develop these models, a reaction scheme, which corresponds to a network of chemical reactions, is postulated and is based on the evolution of observed chemical species and finally, on some theoretical or heuristic considerations of possible reaction pathways [24]. The named power law, is an approach which is frequently used to express chemical reaction rates [25]. Once the structure of the model is established, the calculation of the kinetic parameters of the model is carried out. These are generally determined in two different ways; either one at a time, considering the different components and processes of the model individually, or collectively calibrating the parameters to make the model fit the experimental measurements [26]. In both cases, if a reasonable statistical fit is achieved, the general kinetic model, comprising the initially postulated reaction scheme and the estimated kinetic parameters, is accepted. If not, the kinetic models and/or in the reaction scheme are modified [27]. Bioprocess modeling is classified into two groups: structured and unstructured models.

As is well known, unstructured models describe fermentation processes, considering that microorganisms and/or cells have a fixed and simple composition. However, by idealizing the process conditions, they are not always able to describe the real dynamics. On the other hand, the so-called structured models use a more detailed approach to cell metabolism, aimed at better describing the dynamic behavior of the process [27,28]. Within the structured models we can find the so-called compartmental models. These combine a better description of the behavior of the fermentation that includes the substrate consumption and the generation of the metabolites of interest, with reasonable mathematical complexity and a smaller number of parameters [26,29].

The present work used a power-law kinetic model for the description of bioethanol production in a batch bioreactor by the microorganism *Saccharomyces cerevisiae* as described

in Equations (1)–(4). The model had a phenomenological approach, such that it was based on the biochemical reaction network, so the proposed model was able to predict the bioreactor dynamic behavior under different operation conditions [26].

Proposed reaction rates:

$$\text{Substrate balance } (s) : \dot{s} = -\alpha_1 s x \tag{1}$$

$$\text{Biomass balance } (x) : \dot{x} = \alpha_2 s x - \alpha_5 E_t \tag{2}$$

$$\text{Ethanol balance } (E_t) : \dot{E}_t = \alpha_3 s E_t - \alpha_6 x \tag{3}$$

$$\text{CO}_2 \text{ balance } (CO_2) : \dot{CO}_2 = \alpha_4 s x \tag{4}$$

here $\alpha_1$, $\alpha_2$, $\alpha_3$, $\alpha_4$, $\alpha_5$, and $\alpha_6$ are the kinetic constants for the concentrations of $s$, $x$, $E_t$, and $CO_2$.

To present Equations (1)–(4) in standard state space form, consider the next expression for the state vector:

$$\aleph = [s, x, E_t, CO_2]^T$$

which allows writing of Equations (1)–(4), in a general form, as:

$$\dot{\aleph} = f(\aleph, u); \ \aleph(t_0) = \aleph_0; \ y = [0, 1, 0, 0]^T \aleph \tag{5}$$

where $\aleph \in \mathbb{R}^4$ is the state vector, $u \in \mathbb{R}^m$ is the input, $f$, $y$, and $h$ are vector fields, and $y \in \mathbb{R}^p$ is the output measurement.

The advantages of the model are as follows: (a) The overall strength of the proposed kinetic modeling approach is that it quantitatively takes into account the factors that determine the rate of reactions for the main fermentation metabolites. (b) This kinetic can be applied independent of the microorganism, i.e., for different strains of yeast only a specific parametric identification must be done. (c) The principles of kinetic modeling are applicable to the extracellular environment from the fermentation in batch operation. (d) Thus, the proposed kinetic model can not only describe the reaction rates of substrate and products, but also show that it also has the ability to predict under different initial experimental conditions.

### 2.2.1. Parametric Identification of the Kinetic Model

Within the optimization methods, we can find local, global and hybrid methods. Local methods are robust and easy to implement, generally requiring some kind of parameter initialization, a position in the parameter space from which to start optimization. The initial parameter set is then improved by repeated application of the optimization algorithm; however, this causes the algorithm to converge slower in terms of the number of evaluations of the objective function. Furthermore, it should be taken into account that the objective function normally has several local optima and that the choice of initial values is crucial to finding the global optimum using local methods [19]. On the other hand, global optimization methods search the parameter space more comprehensively; however, a common drawback with these algorithms is a slower convergence rate [29]. Hybrid optimization methods benefit from both the ability of global methods to explore parameter space and the faster rate of convergence of local methods once they approach a local optimum [19].

A hybrid optimization method proposed by Grisales et al. [19] estimates the value of the parameters of Equations (1)–(4). This method has been reported as the best approach to guarantee an optimal estimation of the parameters at a global level.

Parameter estimation problems for nonlinear dynamical systems arise as the minimization of an objective function. Mathematically, the formulation of the objective function is a nonlinear programming problem that consists of finding a vector of parameters to estimate $P$ from the goodness-of-fit of the model with respect to a given set of experimental data. Each designer can propose the objective function based on their needs. This can

be as simple as some error norm between the experimental data and the model predictions [19,29], or more specifically, where some weighting matrix and some restrictions are added path and points of equality and inequality that express additional requirements for the performance of the system [30,31].

The hybrid approach shown in Figure 4 minimizes the error between the values of the experimental data and the simulated values of the model, which we will define as the following objective function *J*, as is usually considered [19,29]:

$$J = \sum_{t=0}^{t_f} \left( \left[ s_{exp} - s_m \right]^2 + \left[ x_{exp} - x_m \right]^2 + \left[ E_{t_{exp}} - E_{t_m} \right]^2 + \left[ CO_{2_{exp}} - CO_{2_m} \right]^2 \right)^{0.5} \quad (6)$$

here $s_{exp}$, $x_{exp}$, $E_{t_{exp}}$, and $CO_{2_{exp}}$ are the experimental data measured in (g/L) and $s_m$, $x_m$, $E_{t_m}$, and $CO_{2_m}$ are the model estimates.

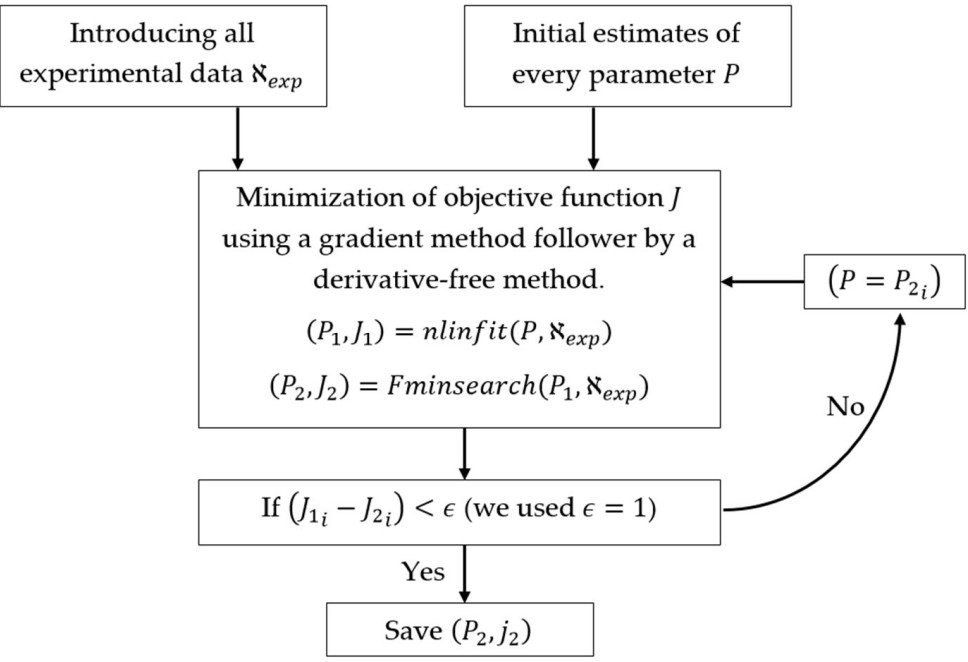

**Figure 4.** In the algorithm used for kinetic fitting, the subscript *i* refers to the value of the functions *J* and *P* at the iteration $i - 1$.

In Figure 4, the first iteration ($P_1$, $J_1$) uses the *nlinfint* library (local method based on the Levenberg–Marquardt algorithm) for the estimations of initial parameters. Later in the iteration ($P_2$, $J_2$) uses the *Fminsearch* method (global method without derivatives) to estimate the optimal values of the parameters. Additionally, if the difference in the minimum value of the objective functions achieved by the two methods is below a threshold $\epsilon$ (in this work $\epsilon = 1$ was used), it is assumed that the resulting parameters in ($P_2$, $J_2$) are the global optimum, if it is not true that ($J_1 - J_2 < \epsilon$) then the first iteration is repeated and $P_2 = P$ is used as the new initial values.

### 2.2.2. Parametric Sensitivity Analysis

A parametric sensitivity analysis determines the degree of change the model undergoes in response to variations of its parameters [30]. This type of analysis is extremely important to determine the region of confidence of the model based on the maximum and minimum values that the kinetic parameters can take. In addition, it allows us to determine the parameters that most influence the dynamics of the model [26].

The Monte Carlo method is a global sensitivity analysis, running parameter sweeps by substituting a range of values (a probability distribution), exploring the design space,

and testing various scenarios. Simulink Design Optimization™ is an interactive Matlab tool that allows you to perform this sensitivity analysis. It has many advantages over local "single point estimation" sensitivity analyses [30]. Among these advantages, the following stand out: (a) graphic results; (b) sensitivity analysis (shows which parameters have the greatest influence on the final results); and (c) scenario analysis. With this method, analysts can see exactly what values each variable has when certain outcomes occur.

### 2.3. Observability

It is important to remember that the implementation of a state observer will only work if, and only if, the system is observable [1]. The concept of observability is useful when solving the problem of estimating non-measurable state variables from measured variables in the shortest possible time. Formally, a system is observable if, for any initial state $\aleph(0)$, (unknown), there exists a finite time $t_1$ such that knowledge of the input $u$ and the output $y$ over the interval $[0, t_1]$ is sufficient to uniquely determine the initial state $\aleph(0)$ [6].

Observability via an Inferential Diagram

This technique studies the observability of nonlinear systems, by exploring the connections between their states, inputs, and outputs [32,33]. The technique is based on the construction of an inference diagram based on the structure of the system model. The inference diagram is built considering the following points [32]:

(a) Draw a bond, $\aleph_i \rightarrow \aleph_j$, *if* $\aleph_j$ appears in the differential equation for $\aleph_i$. This implies that by monitoring $\aleph_i$ it is possible to obtain information about $\aleph_j$.

(b) Decompose the inference diagram into a unique set of maximal strongly connected components (SCC). SCCs are subgraphs selected such that there is a direct path from every node to every other node in the subgraph. Dotted lines enclose the SCCs. It is worth noting that each node in an SCC contains information about the other nodes. The so-called root SCCs do not have output links.

(c) We chose at least one node of each root SCC, which would be the sensor node, to guarantee the observability of the whole system.

This technique explicitly takes advantage of the network structure of the dynamic system [32]. Equation (5) can be structurally represented as a corresponding inference graph $\vartheta$ whose nodes are the internal state variables $\aleph = \{\aleph_1, \ldots, \aleph_n\}$.

- The links in $\vartheta$ capture the pattern of interaction between the state variables: there is a link from $\aleph_j$ to $\aleph_i$ in the graph $\vartheta$ if $\vartheta_{ij}$ is nonzero.
- A node $\aleph_i$ in the graph $\vartheta$ is a sensor node if $y_{ij} \neq 0$ for some $i$.
- A node $\aleph_k$ is an objective node if $F_k \neq 0$ for some $k$.

$$\vartheta = \begin{bmatrix} \star & \star & 0 & 0 \\ \star & \star & \star & 0 \\ \star & \star & \star & 0 \\ \star & \star & 0 & 0 \end{bmatrix}$$
$$y = \begin{bmatrix} 0 & \star & 0 & 0 \end{bmatrix}$$
$$F = \begin{bmatrix} \star & 0 & \star & \star \end{bmatrix} \tag{7}$$

Figure 5 shows the inference diagram of Equation (5), where $\vartheta$ is the matrix of the system, the set of state variables $\aleph = \{\aleph_1, \ldots, \aleph_4\}$ is represented by nodes on the graph, where the sensor nodes $\delta = \{\aleph_2\}$ (defined by $y$) are marked in red and the set of target nodes $\mathcal{O} = \{\aleph_1, \aleph_3, \aleph_4\}$ is marked in green.

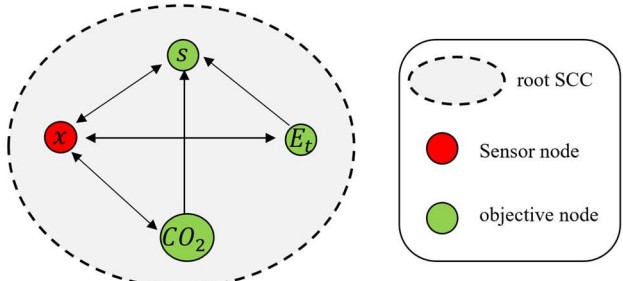

**Figure 5.** Observability analysis (inference diagram) of Equation (5).

**Postulate 1:** *Equation (5) is observable for the output vector* $y = [0, 1, 0, 0]^T$ $\aleph$.

## 2.4. Statistical Correlation Criteria

Before using any model, its statistical quality must be established. This is done by analyzing the deviation between the experimental data and the model results [26]. For a model to be statistically representative, these residuals must be small enough and uncorrelated.

**Coefficient of determination $R^2$:** Describes the proportion of the total variance in the experimental data that can be explained by the model. $R^2 \in (0, 1]$. The model has a good degree of fit to the experimental data when $R^2 \approx 1$.

**Coefficient of efficiency:** Let E be the efficiency coefficient. The proposed model is said to have a good degree of fit with respect to the sample values when $E \in (0, 1]$, while $E > 0.9$ represents a good match between the model and the experimental data [26].

**Relative standard deviation (RSD):** Describes the dispersion of the data with respect to the mean and the result. It is expressed as a percentage, and is particularly useful for comparing the uncertainty between different measurements.

## 2.5. State Observers for Batch Bioreactor Fermentations

A batch fermentation is a closed system for mass transfer. At the beginning of its operation a sterilized nutrient solution is added and inoculated with the microorganism, allowing incubation to take place under optimal fermentation conditions. Fermentation stops at the end of the log phase for primary metabolites, or before the death phase begins for secondary metabolites. The online monitoring of the main variables, i.e., concentrations, and reaction rates, is a problem. On the one hand, we have the high cost of the equipment and the absence of appropriate methods for obtaining the readings, and on the other hand, the risk of contamination of the fermentation at the time of sampling. In this sense, state observers are a viable alternative for determining the variables of the fermentation to face some on-line monitoring problems.

### 2.5.1. Sliding-Mode Observers (SMOs)

Batch bioreactors are highly nonlinear finite-time converging systems, generally said to converge to an equilibrium point. Bioreactors show parametric and modeling uncertainties, related mainly to the kinetic terms mainly. Parametric uncertainties are due to identification issues, environmental effects, and inoculum preparation. On the other hand, modeling uncertainties are generated when the designer ignores part of the system dynamics and makes assumptions such as homogeneity or some other simplifications. In addition, it is important to consider the noise in the measurements [34].

A feature of sliding-mode observers is their low computational effort [35], finite-time convergence, and robustness to modeling uncertainty, perturbations, and measurement noise [8]. Furthermore, SMOs are characterized by their ability to generate a sliding motion on the error between the measured plant output and the observer output, ensuring that the observer produces a set of state estimates that are precisely proportional to the actual output of the plant. When measurement perturbations are present, the sliding-mode observer can

force the estimation errors of the state variables which belong to the observable subspace to converge to zero in finite time. In addition, the disturbances within the system can also be reconstructed [35]. All of these advantages make sliding-mode observers interesting candidates for on-line implementation in a batch bioreactor.

For all of the considered observers, the biomass concentration, which is a feasible concentration measurement is selected as the bioreactor measured output, in order to infer the substrate, ethanol, and carbon dioxide concentrations.

The selected sliding-mode observers are:

Classic Sliding-Mode Observer

For the design of this type of observer, a sliding variable is selected, which represents the difference between the measured variable $(y)$ and the estimated one $(\hat{y})$, so that it has a relative degree 1 with respect to the designed injection signal. The discontinuous control signal acts on the first derivative with respect to the time of the sliding surface $\sigma$ to maintain the trajectories of the system in the sliding set $y - \hat{y} = 0$ [35]. The discontinuous term is the one that allows the system to reject disturbances and parametric uncertainties [35], but it is also the one that produces chattering.

In most cases, sliding-mode observers are obtained by injecting a nonlinear discontinuous term that depends on the output error within the observing system. The discontinuous injection must be designed so that the system trajectories are constrained to lie on some sliding surface in the error space. The resulting movement is called sliding mode [35].

A sliding-mode observer for Equation (5), has the following form:

$$\dot{\hat{\aleph}} = f\left(\hat{\aleph}, u\right) + L sign(y - \hat{y}) \tag{8}$$

where the next expression defines the sign function:

$$sign(y - \hat{y}) = \begin{cases} 1 \ \textbf{\textit{si}} \ y - \hat{y} > 0 \\ 0 \ si \ y - \hat{y} = 0 \\ -1 \ si \ y - \hat{y} < 0 \end{cases} \tag{9}$$

Defining the estimation error as $e = y - \hat{y}$, then, the next equations describe its corresponding dynamics:

$$\dot{e} = \Delta f + \delta f + L sign(e) \tag{10}$$

where $\Delta f = f(\aleph, u) - f\left(\hat{\aleph}, u\right)$ and $\delta f$ is the modeling error. The value of the gain $L$ is assigned based on the next bound [36]:

$$L \geq \eta + F \tag{11}$$

being $\eta$ a positive constant, and $|\Delta f + \delta f| \leq F$.

According to Filippov, the remaining gains can be obtained by applying the concept of equivalent dynamics to the error equations in Equation (10) and linearizing with respect $\hat{\aleph}$, [36].

Proportional Sliding-Mode Observer (PSMO)

Aguilar et al. [8] designed the following observation strategy: To provide robust properties to the observer against disturbances, they considered proportional and sliding mode contributions of the measured error.

This observer structure is related to identification and observation problems by including an uncertainty estimator and a state observer. The observer proportional part has stabilizing effects on the observer performance; high proportional gains ensure that the estimation error will decrease. To guarantee the stabilizing properties, the proportional gains must be in function of a positive solution of the Riccati algebraic equation. The sliding part of the observer serves to compensate for uncertain nonlinear terms and provides

asymptotic convergence. When sufficiently large sliding gains are chosen, the instability effect of the bounded nonlinear element can be decreased. This behavior occurs because, once on the sliding surface, the trajectories of the system remain on that surface, so the sliding condition is taken and the surface and the invariant are configured. This implies that some disturbances or dynamic uncertainties can be compensated for by keeping the surface as an invariant set. For more information check the stability properties and the convergence test in [8].

The following dynamical system is an asymptotic observer of Equation (5):

$$\dot{\hat{x}} = f\left(\hat{x}, u\right) + K(y - \hat{y}) + L sign(y - \hat{y}) \tag{12}$$

Here $L = [L_1, L_2, L_3, L_4]^T$ is the observer gain vector sliding mode and $K = [K_1, K_2, K_3, K_4]^T$ is the gains of the proportional part.

High-Order Sliding-Mode Observer (HOSMO)

The so-called high-order sliding-mode techniques claim to be robust in coping with uncertainties, perturbations, chatter reduction, and finite-time convergence [37]; these techniques have been applied to the design of controllers and design of observers for triangular systems [38,39]. Finite-time observers and controllers have been applied to robot manipulations, and secure data transmission [40].

These algorithms solve the exact stabilization problem in finite time for an output with an arbitrary relative degree. They have proved to be optimal for the estimation of states in systems with the presence of unknown external disturbances. They do not need detailed mathematical models of the plant, and furthermore, they can achieve noise reduction generated by uncertainties in an arbitrary way by artificially increasing the relative degree of the system [35].

The dynamic system described by Equation (13) is a finite-time observer for Equation (5) [41].

$$\dot{\hat{x}} = f\left(\hat{x}, u\right) - L sign(y - \hat{y}) |y - \hat{y}|^{1/p} \tag{13}$$

Here $L$ is the observer gain vector and $p \in Z^+$, where $p > 1$, $p$ is an odd number.

**Remark 1.** *The HOSMO observer is designed to converge in finite time and to improve the rate of convergence, however, it is important to note that high-gain observers tend to amplify measurement noise.*

*2.6. Performance Indexes*

The performance of the observers was evaluated by the following indexes, the IAE (integral absolute error), and the ISE (integral squared error) [42], which are defined below:

$$IAE = \int_0^\infty |e(t)| dt \tag{14}$$

$$ISE = \int_0^\infty e(t)^2 dt \tag{15}$$

The ISE index penalizes the response that has large errors, which usually occurs at the beginning of a response. On the other hand, the IAE is less severe in penalizing a response with large errors and will take into account large and small errors.

**3. Results and Discussion**

*3.1. Bioreactor Performance*

Figure 6 shows the dynamic evolution in the ethanol production process with different initial substrate concentrations of 44 and 54 g/L. The maximum production of $CO_2$ at the end of fermentation was 8.1 g/L and 11.8 g/L, respectively. The maximum ethanol

concentration was 17 g/L for 24 h of operation with an initial substrate concentration of 54 g/L and a biomass concentration of 4.2 g/L at the end of the fermentation. The minimum ethanol concentration was 14.86 g/L for 24 h of operation with an initial substrate concentration of 44 g/L and biomass growth of 3.8 g/L.

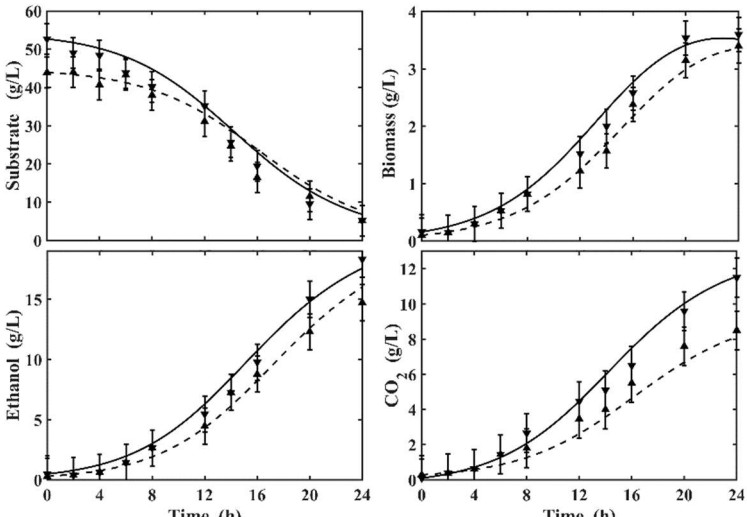

**Figure 6.** Dynamic behavior of the proposed model, experimentally validated for the different initial concentrations of glucose, 54 g/L (solid line) and 44 g/L (dashed line).

The results presented in Figure 6 reveal that the proposed kinetic model was able to predict the concentrations of ethanol, substrate, biomass, and $CO_2$ (lines), obtained from experimental data (symbols). The observed trend indicates that the fermenting microorganism metabolizes the substrate to produce ethanol. This observation confirms the corresponding progressive increase in ethanol and biomass concentrations as cells metabolize the substrate to induce growth and subsequently produce ethanol [26].

The parameters of Equations (1)–(4) were estimated by the hybrid methodology of Section 2.2.1, in Table 2 we show the optimal values of the kinetic parameters. To generate the results reported in this paper, the ODEs were integrated using the ode15s function in MATLAB 2016a®. The MATLAB® functions *fminsearch* (derivative-free method) and *flinfit* (gradient method) were used to minimize the objective function.

**Table 2.** Kinetic parameters under different operating conditions ($s_0 = 54$, $s_0 = 44$).

| Symbol | Value | Units | Definition |
|:---:|:---:|:---:|:---:|
| $\alpha_1$ | $0.048 \pm 0.05$ | L/gh | Substrate kinetic constant |
| $\alpha_2$ | $0.0058 \pm 0.001$ | L/gh | Biomass kinetic constant |
| $\alpha_3$ | $0.0056 \pm 0.01$ | L/gh | Ethanol kinetic constant |
| $\alpha_4$ | $0.0105 \pm 0.001$ | L/gh | $CO_2$ kinetic constant |
| $\alpha_5$ | $0.0075 \pm 0.001$ | $h^{-1}$ | Kinetic constant |
| $\alpha_6$ | $0.0025 \pm 0.001$ | $h^{-1}$ | Kinetic constant |

Figure 7 shows the results of the parametric sensitivity analysis of the model using the Monte Carlo method. According to these results, $\alpha_2$, $\alpha_3$, and $\alpha_5$ were the parameters with the greatest influence on the dynamics of the substrate. The biomass dynamics were more affected by the set of parameters $\alpha_2$, $\alpha_3$, and $\alpha_1$. On the other hand, ethanol production was affected by the parameters $\alpha_3$, $\alpha_2$, and, $\alpha_1$. For the dynamics of $CO_2$ production, the most influential parameters were $\alpha_2$, $\alpha_4$, and, $\alpha_1$.

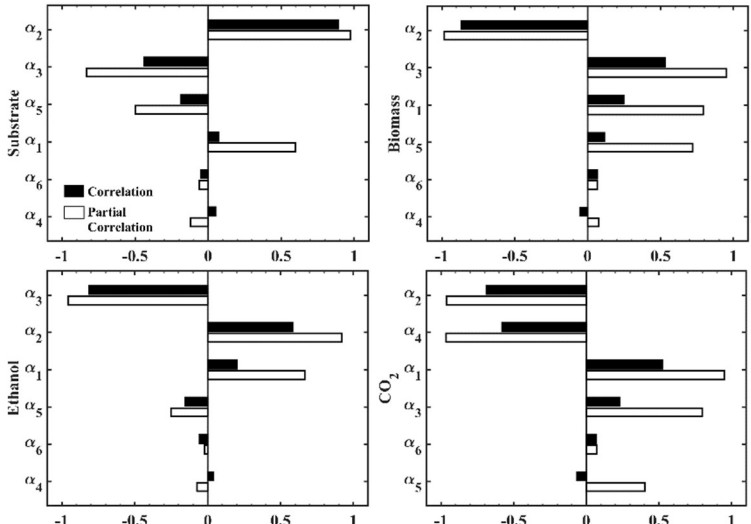

**Figure 7.** Parametric sensitivity analysis of the proposed model.

A common way to assess the fit of the model is to use statistical indicators such as the coefficient of determination ($R^2$, $E$, $RSD$). From this criterion, it was concluded that the model accurately portrayed the experimental data, evaluated by applying $R^2$ (average) = 0.8852 and $E$ (average) = 0.8617 for the two initial conditions. Moreover, in all cases, $R^2$ and $E$ were close to unity, indicating that the model produced a good fit. The relative standard deviation, $RSD$, is used to illustrate the degree of variability of the experimental data set relative to the data set generated by the model. In other words, for small values of $RSD$, the smaller the spread of the data will be [26], as is shown in Table 3.

**Table 3.** Statistical correlation coefficients for quantifying the effectiveness of the model in describing the experimental observations related to discontinuous fermentation.

| Variable | $R^2$ | $E$ | $RSD$ |
|---|---|---|---|
| Initial condition ($s_0 = 54$ g/L, $x_0 = 0.1$ g/L, $Et_0 = 0.29$ g/L, $CO_2 = 0.01$ g/L) | | | |
| Substrate | 0.8932 | 0.8866 | 0.0780 |
| Biomass | 0.8951 | 0.8527 | 0.7233 |
| Ethanol | 0.8811 | 0.8460 | 0.7740 |
| $CO_2$ | 0.8715 | 0.8618 | 0.7498 |
| Initial condition ($s_0 = 44$ g/L, $x_0 = 0.1$ g/L, $Et_0 = 0.29$ g/L, $CO_2 = 0.01$ g/L) | | | |
| Substrate | 0.8854 | 0.8778 | 0.9428 |
| Biomass | 0.8625 | 0.8573 | 0.7628 |
| Ethanol | 0.8651 | 0.8554 | 0.8205 |
| $CO_2$ | 0.8245 | 0.8088 | 0.7972 |

### 3.2. Simulation of Selected Sliding-Mode Observers

The simulation of the selected state observer structures (Table 4) was performed in MATLAB. To generate the results reported in this paper, the ODEs were integrated using the function ode45s, and the observer gains were heuristically tuned.

**Table 4.** Structures of state observers.

| Observer | Structure | Reference |
|---|---|---|
| SMO | $\dot{\hat{x}} = f\left(\hat{x}, u\right) + Lsign(y_{sensor} - \hat{y})$ | [33] |
| PSMO | $\dot{\hat{x}} = f\left(\hat{x}, u\right) + K(y_{sensor} - \hat{y}) + Lsign(y_{sensor} - \hat{y})$ | [8] |
| HOSMO | $\dot{\hat{x}} = f\left(\hat{x}, u\right) + Lsign(y_{sensor} - \hat{y})|y_{sensor} - \hat{y}|^{1/p}$ | [41] |

For the proposed simulations, a copy of Equation (5) was used and white noise was added to emulate the noisy signal of the biomass sensor, where $\delta(t) = rand(1)$.

$$\dot{\aleph} = f(\aleph, u); \quad \aleph(t_0) = \aleph_0$$
$$y = [0, 1, 0, 0]^T \aleph + \delta(t) \tag{16}$$

**Remark 2.** *All observation strategies have the same gain vector L and the structure of each one is expected to influence its performance.*

Figure 8 shows the performance of each of the structures of the observers. Note that they have a good performance and are capable of tracking the trajectories generated by the model.

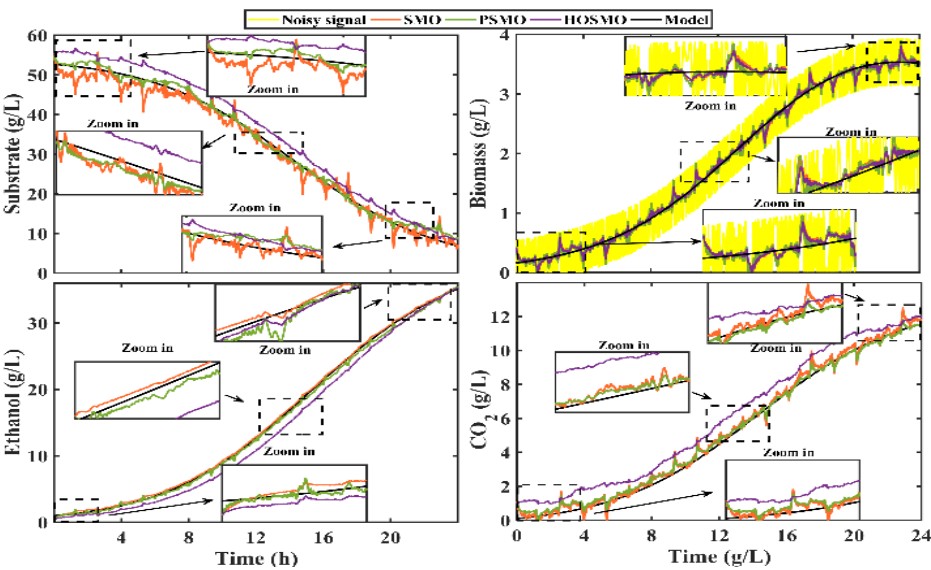

**Figure 8.** Simulation of the observation structures.

The performance of each observer was evaluated through the performance indexes. According to Table 5, the OPSM had a better performance. With the smaller values of IAE and ISE, the proportional gains $(K)$, helped the observer to improve its performance.

**Table 5.** Estimation error index.

| Observer | IAE | | | | ISE | | | |
|---|---|---|---|---|---|---|---|---|
| | $s$ | $x$ | $Et$ | $CO_2$ | $s$ | $x$ | $Et$ | $CO_2$ |
| OSM | 3.889 | 2.556 | 0.988 | 14.39 | 10.83 | 6.44 | 0.187 | 11.61 |
| OPSM | 2.864 | 2.521 | 0.874 | 12.45 | 5.384 | 6.495 | 0.175 | 9.566 |
| OHOSM | 30.27 | 2.614 | 26.32 | 40.99 | 23.23 | 30.4 | 1.026 | 14.99 |

**Remark 3.** *The OPSM exhibits the best performance and allows adequate management of the measurement noises. Furthermore, it is observed that the gain vector K has stabilizing effects on the observer performance and the sliding part of the observer helps to counteract model uncertainties.*

### 3.3. Implementation of the State Observers in Real-Time

The results of the real-time implementation of the state observers are shown in Table 4, however, when programming the observers the biomass sensor signal was used. In this work, a TS-300B turbidity sensor was used to measure biomass density. The sensor is composed of an infrared light emitting diode on one side and a phototransistor to detect

the intensity of the light passing through the open channel to the opposite side [21]. The light intensity on the detection side decreases as the turbidity increases. The output signal ($V_{out}$) is read directly by the NI cRIO 9030.

Figure 9 shows the experimental prototype where the estimation algorithms were run in real-time. The observers were programmed on the NI cRIO 9030 with a 30 ms sample time though the LabVIEW Real-Time [43].

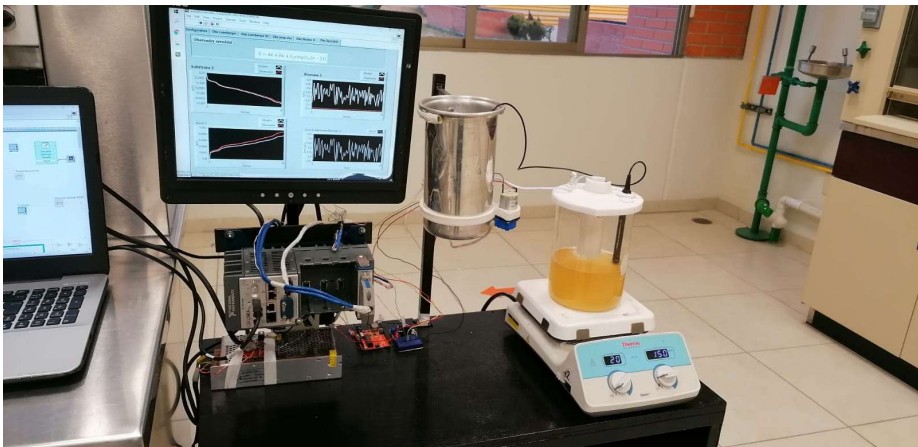

**Figure 9.** Implementation of observation strategies in the prototype plant.

The trajectories of each state observer (solid lines) were extracted from the NI cRIO 9030 and imported into MATLAB. In addition, a comparison with the experimental data was performed. Figure 10 shows the performance of each state observer.

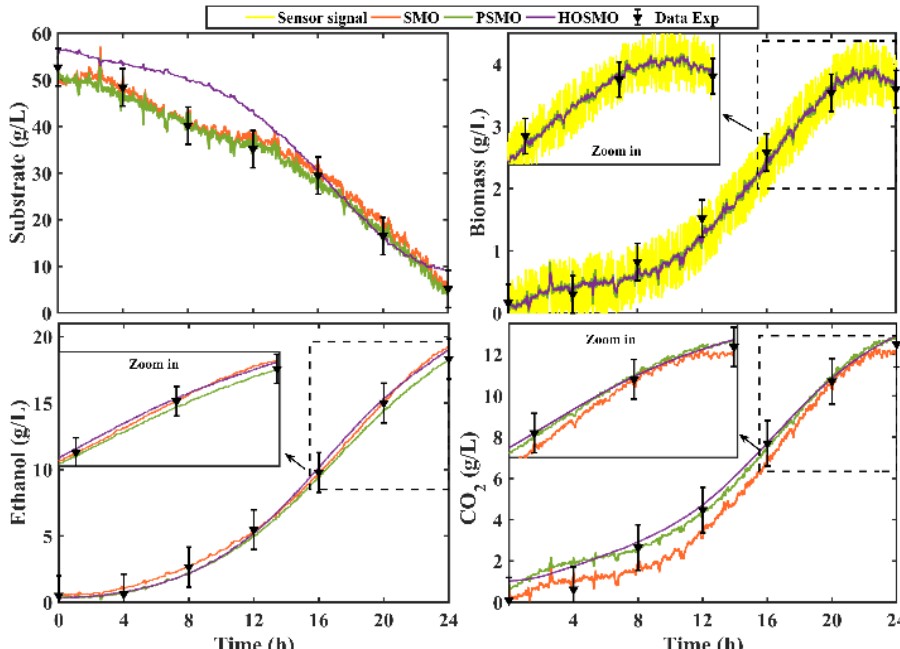

**Figure 10.** Real-time implementation of the observation structures and their comparison with the off-line experimental data.

The corresponding values of the performance indexes IAE and ISE are depicted in Table 6. The OPSM was the observer with the best performance.

**Table 6.** Real-time estimation error.

| Observer | IAE | | | | ISE | | | |
|---|---|---|---|---|---|---|---|---|
| | $s$ | $x$ | $Et$ | $CO_2$ | $s$ | $x$ | $Et$ | $CO_2$ |
| OSM | 2.364 | 0.268 | 0.476 | 1.006 | 6.076 | 0.018 | 0.074 | 0.433 |
| OPSM | 1.495 | 0.266 | 0.392 | 0.689 | 2.791 | 0.017 | 0.058 | 0.248 |
| OHOSM | 27.54 | 0.294 | 1.56 | 2.845 | 13.29 | 0.021 | 0.854 | 3.151 |

Signal Conditioning of the Turbidity Biomass Sensor to Improve the Performance of the PSMO in Real-Time

In most electronic applications, it is natural to try to obtain signals free of uncertainties that could affect the performance of a system. This problem has motivated scientists and engineers to develop algorithms capable of separating components that are mixed and that are capable of rejecting undesirable components. A digital filter is a mathematical operation that takes a sequence of numbers (the input signal) and modifies it by producing another sequence of numbers (the output signal) to enhance or attenuate certain characteristics [44].

In this research work, we used an analog sensor to measure biomass density, which presents high-frequency white noise. Figure 11 shows the behavior of the sensor signal (yellow line). To filter the signal we used a first-order low-pass filter. In this type of filter, the desired cutoff frequency (CF) is established; the filter allows the passage of signals with a lower CF and attenuates signals with frequencies higher than the CF.

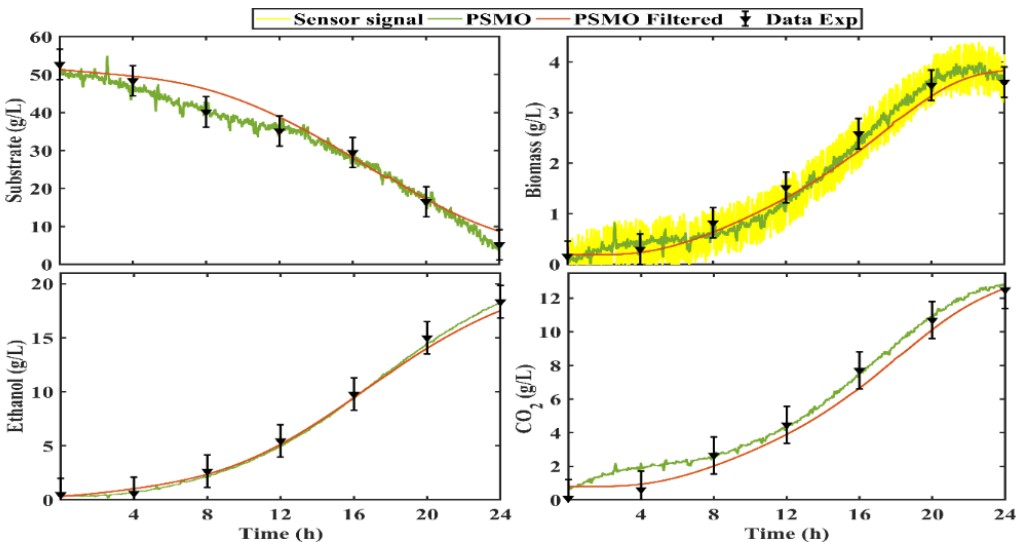

**Figure 11.** PSMO implementation using a filtered sensor signal.

According to the results obtained in Sections 3.2 and 3.3, the PSMO exhibited the best performance.

This section presents the implementation of the PSMO with the filtered signal of the turbidity sensor.

The following differential equation describes the filter [44]:

$$\dot{\varphi}(t) + a\varphi(t) = bu(t) \tag{17}$$

where $u(t)$ and $\varphi(t)$ represent the input and output signals of the filter, $a$ is the cutoff frequency, and $b$ is a constant parameter.

Figure 11 shows the estimates produced by the PSMO without and with the filter (Equation (17)). Note that the filter reduces the effects of measurement noise and that the observer gains have been adjusted for measurement noise. However, the observer had robust properties against disturbances in measurements. Figure 10 graphically confirms

how the PSMO had a good performance, adjusting to the experimental data, although the measurement signal was subjected to a treatment to reduce noise.

## 4. Conclusions

The state observers studied in this work have shown satisfactory performance. Numerical simulations show that the observers are robust to measurement noise. This study also proposes a new power-law kinetic model for the description of fermentative ethanol production in a batch bioreactor by the microorganism *Saccharomyces cerevisiae*. Moreover, a hybrid strategy was implemented for the estimation of the kinetic parameters. An observability analysis based on inference diagrams was performed to determine that the system is fully observable by considering the biomass concentration as measured system output. The state observers studied have shown satisfactory performance. Firstly, numerical simulations show that the observers are robust to measurement noise in accordance with the performance indexes. In addition, the real-time implementation of all the observation strategies was carried out and the performance of each one was evaluated. The sliding-mode proportional observer showed a better performance since its proportional structure helped to attenuate measurement noise, and the sliding part helped to counteract the effects of un-modeled uncertainties. It was also observed that processing the turbidity sensor signal with a low-pass filter improves the rejection of measurement noise and then the performance of the sliding mode proportional observer.

**Author Contributions:** Conceptualization and methodology, E.A.-S., R.A.-L. and P.A.L.-P.; software, E.A.-S., P.A.L.-P. and R.A.-L.; validation, J.L.M.-M., R.A.G.-M., F.P.-G. and E.A.-S.; formal analysis, R.A.-L., R.A.G.-M. and J.L.M.-M.; resources, J.L.M.-M.; writing—original draft preparation, E.A.-S.; writing—review and editing, E.A.-S., R.A.-L., P.A.L.-P., J.L.M.-M., R.A.G.-M. and F.P.-G. All authors provided critical feedback and helped shape the research, analysis, and manuscript. All authors have read and agreed to the published version of the manuscript.

**Funding:** The research for this manuscript was funded by the Secretaria de Investigacion y Posgrado of the Instituto Politécnico Nacional (SIP-IPN) under the research grant SIP20221338.

**Institutional Review Board Statement:** Not applicable.

**Informed Consent Statement:** Not applicable.

**Data Availability Statement:** Not applicable.

**Acknowledgments:** Eduardo Alvarado Santos is a doctoral student from Programa de Doctorado en Ciencias Especialidad en Biotecnología, Centro de Investigación y Estudios Avanzados del Instituto Politécnico Nacional (Cinvestav) and received fellowship 626996 from CONACYT. We thank the Colección Nacional de Cepas Microbianas y Cultivos Celulares CINVESTAV for the *Saccharomyces cerevisiae* strain. We thank the Instituto Politecnico Nacional and Escuela Superior de Apan, Universidad Autónoma del Estado de Hidalgo for supporting this research work.

**Conflicts of Interest:** The authors declare no conflict of interest.

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
