# Peer review of "Comparative Analysis of a Family of Sliding Mode Observers under Real-Time Conditions for the Monitoring in the Bioethanol Production"

_fermentation, doi:10.3390/fermentation8090446_

Round 1

Reviewer 1 Report

The comments are in the attached file.

Reviewer 2 Report

Thanks for the opportunity to review the manuscript by Alvarado-Santos et al. I find the work merits publishing, but I have some concerns.

General concerns:

  1. Please define if the proposed model could be applied to any fermentation process independent of the organism. i.e., bacteria or yeast.
  2. Please compare their model approach versus other model systems used to infer the performance of bioprocesses —for instance, ODE or stochastic modeling.
  3. Please clarify the most critical parameters, variables, or reactions your model retries o simulate in the bioprocess.
  4. Please describe or refers to the geometry measures of the bioreactor used and if your approach can be helpful for the diverse types of bioreactors.

Minor concerns:

  1. Line 148. correct to mg mL-1
  2. Line 151. correct the subindex 2 in CO2-BTA
  3. Lines 201-202, it is duplicated the word "data."
  4. Line 218, there is an unnecessary point after the word "variables"
  5. In figure 6, it is confused which data correspond to 54 and 44 g/L of the substrate (line 338).
  6. Line 339, I think you refer to figure 6, not 4.
  7. Line 350, table 2, it isn't easy to know which data correspond to So=54 or 44.
  8. Reference 39 is missing in the text
  9. You can consider citing to https://doi.org/10.1016/j.conengprac.2013.03.003

Round 2

Reviewer 1 Report

I did a fast forward reading of the responses to the comments, and they seem adequate.

On my part, I agree with the publication of this paper.